# Macrocyclic Ionic Liquids with Amino Acid Residues: Synthesis and Influence of Thiacalix[4]arene Conformation on Thermal Stability

**DOI:** 10.3390/molecules27228006

**Published:** 2022-11-18

**Authors:** Olga Terenteva, Azamat Bikmukhametov, Alexander Gerasimov, Pavel Padnya, Ivan Stoikov

**Affiliations:** 1A.M. Butlerov Chemistry Institute, Kazan Federal University, 18 Kremlevskaya Street, Kazan 420008, Russia; 2Federal Center for Toxicological, Radiation and Biological Safety, 2 Nauchny Gorodok Street, Kazan 420075, Russia

**Keywords:** thiacalix[4]arenes, ionic liquids, amino acid, synthesis, thermal stability, melting point

## Abstract

Novel thiacalix[4]arene based ammonium ionic liquids (ILs) containing amino acid residues (glycine and *L*-phenylalanine) in *cone*, *partial cone*, and *1,3-alternate* conformations were synthesized by alkylation of macrocyclic tertiary amines with *N*-bromoacetyl-amino acids ethyl ester followed by replacing bromide anions with bis(trifluoromethylsulfonyl)imide ions. The melting temperature of the obtained ILs was found in the range of 50–75 °C. The effect of macrocyclic core conformation on the synthesized ILs’ melting points was shown, i.e., the ILs in *partial cone* conformation have the lowest melting points. Thermal stability of the obtained macrocyclic ILs was determined via thermogravimetry and differential scanning calorimetry. The onset of decomposition of the synthesized compounds was established at 305–327 °C. The compounds with *L*-phenylalanine residues are less thermally stable by 3–19 °C than the same glycine-containing derivatives.

## 1. Introduction

Ionic liquids (ILs) have been attracting the attention of researchers in the last two decades. Their unique properties, e.g., low vapor pressure, low toxicity, recyclability, high solvating ability, polarity, thermal and electrochemical stability and electrical conductivity can be explained by the structure [1,2,3,4,5,6,7,8,9,10,11,12,13,14,15,16]. ILs consist of bulky organic cations with low symmetry and inorganic or organic anions [17]. The physicochemical properties of ILs can also be affected by varying the cations and anions [18].

Despite the fact that the first examples of ILs are derivatives of ammonium salts, this class of compounds has not been fully investigated. Ammonium ILs were used for metal ions extraction, functional materials creation, as a reaction medium, battery electrolytes, components of pharmaceutical agents [19,20,21,22,23]. However, low biocompatibility and complexation selectivity limit the practical application of these compounds. A possible solution of this problem is the modification of ILs with various functional groups, e.g., amide, hydroxyl, carboxyl, amino acid fragments etc. ILs with amino acid fragments also increase the stability of biomolecules such as enzymes and DNA [24,25,26,27,28,29]. Prior works showed the rise of thermal stability and effect of such ILs on proteins activity and stability, their package changing and aggregation inhibition [30,31,32,33].

One of the actively developing classes of ILs are macrocyclic and polyionic liquids containing several cationic fragments [34,35,36]. The design of ILs based on supramolecular platforms, e.g., crown ethers [37], pillararenes [38,39,40], and (thia)calixarenes [41,42,43,44,45,46,47], was described. At this moment, the creation of such structures is a non-trivial synthetic problem. Only a few examples of their successful synthesis are known [48,49]. Our scientific group has shown that the introduction of quaternary ammonium, ester and amino acid fragments leads to obtain macrocyclic ILs in *cone* and *1,3-alternate* conformations [50]. There are no literature examples of the synthesis and properties study of macrocyclic ILs based on *partial cone* stereoisomer. The structure of *partial cone* is less symmetrical than *cone* and *1,3-alternate*, that can hypothetically decrease the melting point of such compounds. The introduction of different amino acid residues affects the thermal stability and melting point, which are important characteristics of ILs. In this work, *p-tert*-butylthiacalix[4]arenes tetrasubstituted at the lower rim with quaternary ammonium groups and amino acid fragments (glycine and *L*-phenylalanine) in *cone*, *partial cone*, and *1,3-alternate* conformations were synthesized for the first time. The influence of the conformation of macrocyclic core and the nature of the amino acid substituent on their thermal stability was investigated.

## 2. Results and Discussion

### 2.1. Synthesis of p-tert-butylthiacalix[4]arenes Containing Quaternary Ammonium Groups and Fragments of Glycine and L-phenylalanine

Previously, our scientific group developed an approach to the synthesis of macrocyclic ILs [50]. It consisted in the alkylation of *p-tert*-butylthiacalix[4]arene-based tertiary amines, followed by the replacement of bromide anions. Initially, we synthesized highly reactive alkylating agents containing amino acid residues. Glycine **1** and *L*-phenylalanine **2** were selected to evaluate the effect of planar π-aromatic ring systems on physical properties, e.g., melting point and thermal stability. The target compounds were obtained in two steps (Figure 1). The first step was the synthesis of the amino acid esters **3** and **4**. The second step was the interaction of the obtained compounds **3** and **4** with bromoacetic acid bromide. *N*-Bromoacetyl-amino acid ethyl esters **5** and **6** were obtained in 92 and 90% yield.

The next stage of this work was the study of reaction of the obtained alkylating agents **5** and **6** with tetrasubstituted thiacalixarenes **7**–**9** containing terminal tertiary amino groups in *cone*, *partial cone* and *1,3-alternate* conformations (Figure 2). Targeted bromides of macrocyclic quaternary ammonium salts **10**–**15** were obtained in high yields (93–95%). Previously, a significant decrease in the melting point by replacing halide ions with bis(trifluoromethylsulfonyl)imide ions (N(SO_2_CF_3_)_2_^−^_,_ NTf_2_^−^) has also been shown [51]. This can be explained by the fact that the increase of anion size decreases symmetry of the molecule. Thus, the compounds **10**–**15** were reacted with lithium bis(trifluoromethylsulfonyl)imide in water at room temperature. The macrocyclic ILs **16**–**21** were obtained in yields close to quantitative (Figure 2).

The structure and the composition of all synthesized compounds were confirmed by ^1^H and ^13^C NMR, IR spectroscopy, mass spectrometry and elemental analysis (Appendix A). The conformation of modified *p-tert*-butylthiacalix[4]arene derivatives can be determined by one-dimensional ^1^H NMR spectroscopy based on specific proton signals of *tert*-butyl group, aromatic ring, oxymethylene and amide group. Table 1 lists the values of the characteristic chemical shifts of the compounds **10**–**21**. The protons of oxymethylene and amide groups of the compound **15** (*1,3-alternate*) are located in the shielded zone of neighboring aromatic rings of the macrocycle. These signals in the ^1^H NMR spectrum are recorded upfield (4.00 and 8.02 ppm, respectively) of those of the macrocycles **11** in *cone* conformation (4.81 and 8.50 ppm, respectively). The chemical shifts of the aromatic protons depend less on the conformation of the macrocyclic platform, shifting by only 0.20 ppm upfield from *cone* **11** (7.39 ppm) to *1,3-alternate* **15** (7.59 ppm) stereoisomers. This provides evidence of the shielding effect of neighboring aryl fragments in *cone* stereoisomer on the aryl protons of macrocycle ring. The *tert*-butyl groups proton signals of *cone* **11** were found upfield (1.07 ppm) in contrast to the same signals of *1,3-alternate* **15** (1.19 ppm). This effect was related to the spatial location of the *tert*-butyl groups of *1,3-alternate* stereoisomer shielded by neighboring fragments of the macrocycle. The proton signals in *partial cone* **13** differ from *cone* and *1,3-alternate* due to the asymmetric structure of the macrocycle. The *tert*-butyl groups proton signals of *partial cone* **13** were recorded upfield (1.00, 1.10 and 1.27 ppm) as singlets with 2:1:1 intensity ratio. The oxymethylene proton signals were located in 4.40–4.77 ppm as two singlets and an AB–system. The aromatic ring proton signals are recorded at 7.67, 7.75, 7.01–7.65 ppm as two singlets and an AB–system. The amide group proton signals appeared in 8.31–8.48 ppm as broadened triplets. This effect of AB–systems in the ^1^H NMR spectra due to the asymmetry of *partial cone* stereoisomer structure shielded by neighboring aromatic fragments of the macrocycle.

It should be noted that the proton signals in the ^1^H NMR spectra of the compounds **10**–**15** containing halide anions, and the proton signals of salts **16**–**21** containing NTf_2_^−^ anions have identical multiplicity and exert very similar chemical shifts. This can be explained by the ability of compounds to form solvent-separated ion pairs. The quartet observed in the ^13^C NMR spectra of compounds **16**–**21** at 120 ppm corresponds to the N(SO_2_CF_3_)_2_^−^ anion. The obtained salts **10**–**21** were characterized by ESI mass spectrometry. The mass spectra of the compounds **10**–**15** showed peaks corresponding to one-, two-, three-, and four-charged molecular ions without one, two, three, and four bromide anions. The obtained data also confirm the formation of solvate-separated ion pairs by the compounds **10**–**21**.

### 2.2. The Study of Thermal Stability of the Obtained Thiacalix[4]arene Based ILs

Melting point is one of the most important characteristics of ILs. Melting points of the synthesized thiacalix[4]arenes **10**–**21** are presented in Table 2. The replacement of halide ions by NTf_2_^−^ ions leads to significant decrease of the melting points of the thiacalix[4]arenes by 39–55 °C. All synthesized macrocycles **16**–**21** containing NTf_2_^−^ anions melt below 100 °C. It is well known that molecular packing density in the crystal lattice is a major factor affecting the melting point of the compound. More symmetrical molecules have denser packing in crystal and higher melting points. A comparison with the obtained results from previously published compounds with glycine residues [50] found that symmetrical *1,3-alternate* and *cone* stereoisomers showed higher melting points than asymmetrical *partial cone* structures. Thus, our hypothesis of lowering the melting point of *partial cone* stereoisomers (the compounds **12**, **13**, **18**, **19**) due to their molecular asymmetry was confirmed experimentally. However, the decrease of the melting point of the targeted compounds due to aromatic fragments in amino acid residues was not confirmed. The melting points of the macrocycles containing glycine fragments are lower by 1–11 °C compared to stereoisomers with *L*-phenylalanine fragments. Apparently, these results can be explained by the interaction of *L*-phenylalanine fragments with each other and the formation of additional hydrophobic and π-π interactions, which leads to a denser molecular packing in the crystal lattice and an increase in the melting point of the compounds as a result.

High thermal stability is one of the characteristic properties of ILs [52,53]. The correlation between the obtained macrocyclic ILs structure and their thermal stability (influence of macrocycle conformation and amino acid residues) was investigated via thermogravimetric analysis (TG). Figure 1 shows the TG curves for the compounds **16**–**21**. All obtained macrocyclic ILs were thermally stable (decomposition temperature T_onset_ = 305–327 °C). Glycine containing compounds decomposed at a higher temperature (by 3–19 °C) compared to the compounds containing *L*-phenylalanine fragments. Many low molecular weight compounds for biofuel technology are obtained by pyrolysis (thermal lysis) of oligo- and polypeptides at a temperature of 300–350 °C [54]. However, thermal decomposition of proteins occurs at temperatures between 175 and 250 °C. In our case, the decomposition temperature of the amino acids containing compounds **16**–**21** was significantly higher. The obtained results are also consistent with the literature data on the thermal stability of macrocyclic ILs [38,50]. Temperature data T_10%_ and T_50%_ corresponding to 10% and 50% weight loss on decomposition are presented in Table 3. These characteristics are important in materials thermal stability research. The difference between T_onset_, T_10%_ and T_50%_ is the measure of decomposition rate [52,55]. T_onset_ and T_10%_ of the studied compounds differ by 0–3 °C. The difference between T_onset_ and T_50%_ is more considerable, namely 41–109 °C. These results correspond to decomposition rate of non-macrocyclic ILs containing quaternary ammonium fragments [56,57]. The differential scanning calorimetry (DSC) heating curves of the synthesized compounds (Appendix A) in the temperature range of 300–375 °C clearly show endo effects corresponding to the first stage of decomposition. The values of endo effects for the compounds **16**, **18**, **20** are similar (38–41 J/g) (Appendix A). These values are larger for the compounds **17**, **19**, **21** (53–55 J/g) (Appendix A). The further decomposition of the obtained compounds at temperatures above 375 °C is accompanied by exo effects.

The experimental results and determined correlations between the synthesized macrocyclic ILs structure and their thermal characteristics showed that the introduction of aromatic amino acid (*L*-phenylalanine) fragments into ammonium salts based on *p-tert*-butylthiacalixarene reduces the thermal stability by 3–19 °C compared to glycine containing compounds. The melting points of the *L*-phenylalanine derivatives were higher than glycine ones by 1–11 °C. Oligo- and polypeptides thermal stability literature data [58,59,60] show that protein thermal stability increase is associated with rise of the number of charged amino acid fragments in their structures capable of electrostatic and cation–π interactions [61]. The presence of amino acids with uncharged polar fragments in protein structures reduces their thermal stability due to biomolecule packing efficiency decrease [62]. Thus, the obtained macrocyclic ILs are considered as biomimetic models of oligo- and polypeptides with the same structural patterns. The obtained results can also be applied to design of sensor systems capable for target substrate recognition.

## 3. Materials and Methods

### 3.1. General

All chemicals were purchased from Acros (Fair Lawn, NJ, USA), and most of them were used as received without additional purification. Organic solvents were purified by standard procedures. ^1^H NMR and ^13^C NMR spectra were obtained on the Bruker Avance-400 spectrometer (Bruker Corp., Billerica, MA, USA) (^13^C{^1^H} 100 MHz and ^1^H 400 MHz). Chemical shifts were determined against the signals of residual protons of deuterated solvent (DMSO-*d_6_*). The compounds concentration was equal to 3–5% by the weight in all records. The FTIR ATR spectra were recorded on the Spectrum 400 FT-IR spectrometer (Perkin–Elmer, Seer Green, Llantrisant, UK) with the Diamond KRS-5 attenuated total internal reflectance attachment (resolution 0.5 cm^−1^, accumulation of 64 scans, recording time 16 s in the wavelength range 400–4000 cm^−1^). Elemental analysis was performed on the Perkin–Elmer 2400 Series II instruments (Perkin–Elmer, Waltham, MA, USA). Melting points were determined using the Boetius Block apparatus (VEB Kombinat Nagema, Radebeul, Germany). Mass spectra (ESI) were recorded on an AmaZonX mass spectrometer (Bruker Daltonik GmbH, Bremen, Germany). The drying gas was nitrogen at 300 °C. The capillary voltage was 4.5 kV. The samples were dissolved in acetonitrile (concentration ~ 10^−6^ g mL^−1^). ESI HRMS experiments were performed at Agilent 6550 iFunnel Q-TOF LC/MS (Agilent Technologies, Santa Clara, CA, USA), equipped with Agilent 1290 Infinity II LC. Simultaneous thermogravimetry (TG) and differential scanning calorimetry (DSC) of solid samples were performed using the thermoanalyzer STA 449F1 Jupiter (Netzsch, Germany) at the temperature range of 40–500 °C. The measurements were carried out in aluminum crucibles in a dynamic argon atmosphere (75 mL/min) at a temperature scanning rate of 10 °C/min. The weights of sample were 4.9–10.2 mg.

*N*-Bromoacetyl-glycine ethyl ester **5** and thiacalix[4]arenes **7**–**10**, **14**, **16**, **20** were synthesized according to the literature procedures [50,63,64,65].

### 3.2. Procedure for the Synthesis of the Compound ***4***

*L*–Phenylalanine **2** (1 g, 6.05 mmol) was dissolved in 10 mL of ethanol in a round-bottom flask equipped with a magnetic stirrer. Thionyl chloride (0.88 mL, 12.10 mmol) was added dropwise with stirring. The reaction mixture was left for 30 min at room temperature, then the reaction was carried out under heating for 4 h. The solvent was removed on a rotary evaporator. Diethyl ether was added to the residue, after which the formed precipitate was filtered off. The obtained product was dried in vacuum over phosphorus pentoxide.

#### L–Phenylalanine Ethyl Ester Hydrochloride (**4**) [66]

^1^H NMR (DMSO-*d_6_*, δ, ppm, *J*/Hz): 1.07 (t, ^3^*J*_HH_ = 7.1 Hz, 3H, CH_3_CH_2_O), 3.01–3.20 (m, 2H, CH_2_Ph), 4.08 (m, 2H, CH_3_CH_2_O), 4.23 (m, 1H, NHCHCO), 7.22–7.35 (m, 5H, PhCH_2_), 8.64 (br.s, 3H, NH_3_^+^).

### 3.3. Procedure for the Synthesis of the Compound ***6***

The solution of Na_2_CO_3_ (3.46 g, 32.66 mmol) in 50 mL of water was added to the suspension of *L*-phenylalanine ethyl ester hydrochloride **4** (3.41 g, 14.85 mmol) in benzene (50 mL). The reaction mixture was then cooled to 0 °C and bromoacetyl bromide (2.6 mL, 30 mmol) was added dropwise. The reaction mixture was allowed to warm to room temperature and stirred for 12 h with controlling pH = 6.5 by adding acetic acid. The organic phase was separated on a separating funnel and dried over anhydrous magnesium sulfate. Then the benzene was removed on a rotary evaporator. The obtained product **6** was dried in vacuum over phosphorus oxide.

#### *N*-Bromoacetyl-*L*-Phenylalanine Ethyl Ester (**6**) [66]

^1^H NMR (DMSO-*d_6_*, δ, ppm, *J*/Hz): 1.11 (t, ^3^*J*_HH_ = 7.1 Hz, 3H, CH_3_CH_2_O), 2.90–3.05 (m, 2H, CH_2_Ph), 3.86 (s, 2H, BrCH_2_CO), 4.06 (q, ^3^*J*_HH_ = 7.1 Hz, 2H, CH_3_CH_2_O), 4.45 (m, 1H, NHCHCO), 7.20–7.30 (m, 5H, PhCH_2_), 8.75 (d, ^3^*J*_HH_ = 7.5 Hz, 1H, CONHCH).

### 3.4. General Procedure for the Synthesis of the Compounds ***10***–***15***

The compounds **7**–**9** (0.30 g, 0.023 mmol) were dissolved in 5 mL of acetonitrile in a round-bottom flask equipped with a magnetic stirrer and reflux condenser. An equimolar amount per functional group (0.092 mmol) of the alkylating agent (*N*-bromoacetyl-glycine ethyl ester **5** or *N*-bromoacetyl-*L*-phenylalanine ethyl ester **6**) was added. The reaction mixture was refluxed for 18 h. Then the solvent was removed on a rotary evaporator. The obtained product was dried in vacuum over phosphorus oxide.

#### 3.4.1. 5,11,17,23-Tetra-*tert*-butyl-25,26,27,28-tetrakis{*N*-[3′-(dimethyl{[(*S*)-ethoxycarbonylbenzylmethyl]aminocarbonylmethyl}ammonio)propyl]aminocarbonylmethoxy}-2,8,14,20-thiacalix[4]arene Tetrabromide in *cone* Conformation (**11**)

Yield: 0.57 g (96%). M.p. 118 °C. ^1^H NMR (DMSO-*d_6_*, δ, ppm, *J*/Hz): 1.06 (s, 36H, (CH_3_)_3_C), 1.11 (t, ^3^*J*_HH_ = 7.1 Hz, 12H, CH_3_CH_2_O), 1.88 (m, 8H, NHCH_2_CH_2_CH_2_N^+^), 2.93 (m, 8H, CH_2_Ph), 3.08 (s, 24H, (CH_3_)_2_N^+^), 3.20 (m, 8H, NHCH_2_CH_2_CH_2_N^+^), 3.45 (m, 8H, NHCH_2_CH_2_CH_2_N^+^), 4.02–4.13 (m, 16H, CH_3_CH_2_O, N^+^CH_2_CO), 4.55 (m, 4H, NHCHCO), 4.80 (s, 8H, OCH_2_CO), 7.21–7.30 (m, 20H, Ph), 7.38 (s, 8H, ArH), 8.51 (br.s, 4H, NHCH_2_CH_2_CH_2_N^+^), 9.12. (d, ^3^*J*_HH_ = 7.5 Hz, 4H, CONHCH). ^13^C NMR (DMSO-*d_6_*, δ, ppm): 13.9, 22.6, 30.7, 33.9, 35.4, 36.5, 51.1, 53.7, 60.9, 61.7, 62.8, 74.2, 126.7, 128.1, 128.3, 129.1, 134.4, 136.6, 146.7, 157.9, 163.1, 168.3, 170.6. Elemental analysis. C_120_H_168_Br_4_N_12_O_20_S_4_ C, 56.60; H, 6.65; Br, 12.55; N, 6.60; S, 5.04. Found: C, 56.72; H, 6.85; Br, 12.23; N, 6.43; S, 4.79. MS (ESI), *m/z*: calculated for 556.5 [M–4 Br^−^]^4+^, 1192.5 [M–2 Br^−^]^2+^; found: 556.6 [M–4 Br^−^]^4+^, 1193.0 [M–2 Br^−^]^2+^. FTIR ATR (ν, cm^−1^): 1095 (COC), 1677 (C=O), 3191 (N–H).

#### 3.4.2. 5,11,17,23-Tetra-*tert*-butyl-25,26,27,28-tetrakis{*N*-[3′-(dimethyl{[ethoxycarbonylmethyl]aminocarbonylmethyl}ammonio)propyl]aminocarbonylmethoxy}-2,8,14,20-tetrathiacalix[4]arene Tetrabromide in *partial cone* Conformation (**12**)

Yield: 0.477 g (94%). M.p. 105 °C. ^1^H NMR (DMSO-*d_6_*, δ, ppm, *J*/Hz): 1.00 (s, 18H, (CH_3_)_3_C), 1.14–1.21 (m, 12H, CH_3_CH_2_O), 1.27 (s, 9H, (CH_3_)_3_C), 1.29 (s, 9H, (CH_3_)_3_C), 1.94 (m, 8H, NHCH_2_CH_2_CH_2_N^+^), 3.13 (s, 6H, (CH_3_)_2_N^+^), 3.19 (s, 18H, (CH_3_)_2_N^+^), 3.39 (m, 8H, NHCH_2_CH_2_CH_2_N^+^), 3.53 (m, 8H, NHCH_2_CH_2_CH_2_N^+^), 3.93 (d, ^3^*J*_HH_ = 5.8 Hz, 8H, NHCH_2_CO), 4.07–4.13 (m, 16H, CH_3_CH_2_O, N^+^CH_2_CO), 4.40 (d, ^2^*J*_HH_ = 13.6 Hz, 2H, OCH_2_C(O)), 4.49 (s, 2H, OCH_2_C(O)), 4.51 (s, 2H, OCH_2_C(O)), 4.80 (d, ^2^*J*_HH_ = 13.6 Hz, 2H, OCH_2_C(O)), 7.01 (d, ^4^*J*_HH_ = 2.4 Hz, 2H, ArH), 7.65 (d, ^4^*J*_HH_ = 2.4 Hz, 2H, ArH), 7.67 (s, 2H, ArH), 7.75 (s, 2H, ArH), 8.32 (br.s, 2H, NHCH_2_CH_2_CH_2_N^+^), 8.42 (br.s, 1H, NHCH_2_CH_2_CH_2_N^+^), 8.50 (br.s, 1H, NHCH_2_CH_2_CH_2_N^+^), 9.05. (m, 4H, CONHCH). ^13^C NMR (DMSO-*d_6_*, δ, ppm): 14.1, 22.6, 30.7, 31.0, 33.8, 35.5, 40.8, 51.3, 60.8, 61.8, 62.7, 72.6, 126.4, 127.1, 127.6, 128.1, 133.7, 134.0, 135.1, 135.4, 145.3, 145.7, 146.5, 157.2, 159.4, 163.7, 166.9, 168.0, 168.8, 169.1. Elemental analysis. C_92_H_144_Br_4_N_12_O_20_S_4_ C, 50.55; H, 6.64; Br, 14.62; N, 7.69; S, 5.87; Found: C, 51.52; H, 6.25; Br, 14.26; N, 7.47; S, 5.89. HRMS (ESI), *m/z*: calculated for: 466.2370 [M–4 Br^−^]^4+^, 647.9557 [M–3 Br^−^]^3+^, 1012.3919 [M–2 Br^−^]^2+^; found: 466.2364 [M–4 Br^−^]^4+^, 647.9541 [M–3 Br^−^]^3+^, 1012.3937 [M–2 Br^−^]^2+^. FTIR ATR (ν, cm^−1^): 1094 (COC), 1675 (C=O), 3207 (N–H).

#### 3.4.3. 5,11,17,23-Tetra-*tert*-butyl-25,26,27,28-tetrakis{*N*-[3′-(dimethyl{[(*S*)-ethoxycarbonylbenzylmethyl]aminocarbonylmethyl}ammonio)propyl]aminocarbonylmethoxy}-2,8,14,20-thiacalix[4]arene Tetrabromide in *partial cone* Conformation (**13**)

Yield: 0.551 g (93%). M.p. 110 °C. ^1^H NMR (DMSO-*d_6_*, δ, ppm, *J*/Hz): 1.00 (s, 18H, (CH_3_)_3_C), 1.10 (t, ^3^*J*_HH_ = 7.1 Hz, 12H, CH_3_CH_2_O), 1.27 (s, 9H, (CH_3_)_3_C), 1.28 (s, 9H, (CH_3_)_3_C), 1.85 (m, 8H, NHCH_2_CH_2_CH_2_N^+^), 2.90–2.93 (m, 8H, CH_2_Ph), 3.03 (s, 6H, (CH_3_)_2_N^+^), 3.09 (s, 18H, (CH_3_)_2_N^+^), 3.13–3.24 (m, 8H, NHCH_2_CH_2_CH_2_N^+^), 3.46 (m, 8H, NHCH_2_CH_2_CH_2_N^+^), 4.01–4.13 (m, 16H, CH_3_CH_2_O, N^+^CH_2_CO), 4.42 (d, ^2^*J*_HH_ = 13.5 Hz, 2H, OCH_2_C(O)), 4.48 (s, 2H, OCH_2_C(O)), 4.51 (s, 2H, OCH_2_C(O)), 4.55–4.60 (m, 4H, NHCHCO), 4.79 (d, ^2^*J*_HH_ = 13.5 Hz, 2H, OCH_2_C(O)), 7.01 (d, ^4^*J*_HH_ = 2.4 Hz, 2H, Ar-H), 7.22–7.30 (m, 20H, Ph), 7.65 (d, 2H, ^4^*J*_HH_ = 2.4 Hz, ArH), 7.67 (s, 2H, ArH), 7.75 (s, 2H, ArH), 8.31 (br.s, 2H, NHCH_2_CH_2_CH_2_N^+^), 8.41 (br.s, 1H, NHCH_2_CH_2_CH_2_N^+^), 8.48 (br.s, 1H, NHCH_2_CH_2_CH_2_N^+^), 9.12 (d, ^3^*J*_HH_ = 7.5 Hz, 4H, CONHCH). ^13^C NMR (DMSO-*d_6_*, δ, ppm): 14.0, 22.6, 28.8, 30.7, 31.0, 33.8, 34.0, 35.5, 36.5, 51.2, 53.7, 54.0, 60.7, 61.0, 61.6, 61.8, 62.8, 63.1, 72.6, 126.4, 126.7, 127.1, 127.6, 128.3, 129.2, 133.7, 134.1, 135.1, 135.4, 136.6, 145.3, 145.7, 146.5, 157.2, 159.4, 163.2, 166.9, 168.0, 168.8, 170.7. Elemental analysis. C_120_H_168_Br_4_N_12_O_20_S_4_ C, 56.60; H, 6.65; Br, 12.55; N, 6.60; S, 5.04; found: C, 56.52; H, 6.55; Br, 12.26; N, 6.47; S, 4.89. HRMS (ESI), *m/z*: calculated for: 556.5348 [M–4 Br^−^]^4+^, 768.3527 [M–3 Br^−^]^3+^, 1192.4858 [M–2 Br^−^]^2+^; found: 556.5339 [M–4 Br^−^]^4+^, 768.3522 [M–3 Br^−^]^3+^, 1192.4872 [M–2 Br^−^]^2+^. FTIR ATR (ν, cm^−1^): 1094 (COC), 1672 (C=O), 3187 (N–H).

#### 3.4.4. 5,11,17,23-Tetra-*tert*-butyl-25,26,27,28-tetrakis{*N*-[3′-(dimethyl{[(*S*)-ethoxycarbonylbenzylmethyl]aminocarbonylmethyl}ammonio)propyl]aminocarbonylmethoxy}-2,8,14,20-thiacalix[4]arene tetrabromide in *1,3-alternate* Conformation (**15**)

Yield: 0.574 g (97%). M.p. 123 °C. ^1^H NMR (DMSO-*d_6_*, δ, ppm, *J*/Hz): 1.12 (t, ^3^*J*_HH_ = 7.1 Hz, 12H, CH_3_CH_2_O), 1.19 (s, 36H, (CH_3_)_3_C), 1.90 (m, 8H, NHCH_2_CH_2_CH_2_N^+^), 3.01 (m, 8H, CH_2_Ph), 3.09 (s, 24H, (CH_3_)_2_N^+^), 3.15 (m, 8H, NHCH_2_CH_2_CH_2_N^+^), 3.45 (m, 8H, NHCH_2_CH_2_CH_2_N^+^), 4.00 (s, 8H, OCH_2_CO), 4.01–4.14 (m, 16H, CH_3_CH_2_O, N^+^CH_2_CO), 4.59 (m, 4H, NHCHCO), 7.20–7.30 (m, 20H, Ph), 7.59 (s, 8H, ArH), 8.02 (br.s, 4H, NHCH_2_CH_2_CH_2_N^+^), 9.12 (d, ^3^*J*_HH_ = 7.6 Hz, 4H, CONHCH).^13^C NMR (DMSO-*d_6_*, δ, ppm): 13.9, 22.6, 30.7, 33.9, 35.4, 36.5, 51.1, 53.7, 60.9, 61.7, 62.8, 74.2, 126.7, 128.1, 128.3, 129.1, 134.5, 136.6, 146.7, 157.9, 163.1, 168.3, 170.6. Elemental analysis. C_120_H_168_Br_4_N_12_O_20_S_4_ C, 56.60; H, 6.65; Br, 12.55; N, 6.60; S, 5.04; found: C, 56.78; H, 6.26; Br, 12.21; N, 6.31; S, 4.07. MS (ESI), *m/z*: calculated: 556.5 [M–4 Br^−^]^4+^, 768.3 [M–3 Br^−^]^3+^, 1192.5 [M–2 Br^−^]^2+^, 2463.9 [M–Br^−^]^+^; found: 556.5 [M–4 Br^−^]^4+^, 768.9 [M–3 Br^−^]^3+^, 1193.5 [M–2 Br^−^]^2+^, 2464.0 [M–Br^−^]^+^. FTIR ATR (ν, cm^−1^): 1086 (COC), 1675 (C=O), 3186 (N–H).

### 3.5. General Procedure for the Synthesis of the Compounds ***16***–***21***

The compounds **10**–**15** (0.10 g) were dissolved in 2 mL of water in a round-bottom flask equipped with a magnetic stirrer and reflux condenser. An equimolar amount per functional group of the lithium bis(trifluoromethylsulfonyl)imide was added. The reaction mixture was stirred for 24 h. The resulting precipitate was filtered off. The obtained product was dried in vacuum over phosphorus oxide.

#### 3.5.1. 5,11,17,23-Tetra-*tert*-butyl-25,26,27,28-tetrakis{*N*-[3′-(dimethyl{[(*S*)-ethoxycarbonylbenzylmethyl]aminocarbonylmethyl}ammonio)propyl]aminocarbonylmethoxy}-2,8,14,20-thiacalix[4]arene tetra[bis(trifluoromethylsulfonyl)imide] in *cone* Conformation (**17**)

Yield: 0.127 g (97%). M.p. 64 °C. ^1^H NMR (DMSO-*d_6_*, δ, ppm, *J*/Hz): 1.06 (s, 36H, (CH_3_)_3_C), 1.12 (t, ^3^*J*_HH_ = 7.1 Hz, 12H, CH_3_CH_2_O), 1.86 (m, 8H, NHCH_2_CH_2_CH_2_N^+^), 2.89–295 (m, 8H, CH_2_Ph), 3.06 (s, 24H, (CH_3_)_2_N^+^), 3.20 (m, 8H, NHCH_2_CH_2_CH_2_N^+^), 3.42 (m, 8H, NHCH_2_CH_2_CH_2_N^+^), 3.98–4.09 (m, 16H, CH_3_CH_2_O, N^+^CH_2_CO), 4.57 (m, 4H, NHCHCO), 4.79 (s, 8H, OCH_2_CO), 7.19–7.29 (m, 20H, Ph), 7.38 (s, 8H, ArH), 8.48 (br.s, 4H, NHCH_2_CH_2_CH_2_N^+^), 9.06 (d, ^3^*J*_HH_ = 7.5 Hz, 4H, CONHCH). ^13^C NMR (DMSO-*d_6_*, δ, ppm): 13.9, 22.6, 30.7, 33.9, 35.4, 36.6, 51.2, 53.5, 61.0, 61.7, 62.6, 74.2, 119.5 (^1^*J*_CF_ = 322 Hz), 126.8, 128.1, 128.4, 129.1, 134.5, 136.5, 146.8, 158.0, 163.1, 168.3, 170.6. Elemental analysis. C_128_H_168_F_24_N_16_O_36_S_12_ C, 45.93; H, 5.06; F, 13.62; N, 6.69; S, 11.49; found: C, 45.72; H, 5.85; F, 13.23; N, 6.43; S, 11.79. HRMS (ESI), *m/z*: calculated for: 556.5348 [M–4 NTf_2_^−^]^4+^, 835.3524 [M–3 NTf_2_^−^]^3+^; found: 556.5220 [M–4 NTf_2_^−^]^4+^, 835.3328 [M–3 NTf_2_^−^]^3+^. FTIR ATR (ν, cm^−1^): 1094 (COC), 1668 (C=O), 3065 (N–H).

#### 3.5.2. 5,11,17,23-Tetra-*tert*-butyl-25,26,27,28-tetrakis{*N*-[3′-(dimethyl{[ethoxycarbonylmethyl]aminocarbonylmethyl}ammonio)propyl]aminocarbonylmethoxy}-2,8,14,20-tetrathiacalix[4]arene tetra[bis(trifluoromethylsulfonyl)imide] in *partial cone* Conformation (**18**)

Yield: 0.131 g (96%). M.p. 50 °C. ^1^H NMR (DMSO-*d_6_*, δ, ppm, *J*/Hz): 1.00 (s, 18H, (CH_3_)_3_C), 1.14–1.20 (m, 12H, CH_3_CH_2_O), 1.27 (s, 9H, (CH_3_)_3_C), 1.30 (s, 9H, (CH_3_)_3_C), 1.93 (m, 8H, NHCH_2_CH_2_CH_2_N^+^), 3.13 (s, 6H, (CH_3_)_2_N^+^), 3.18 (s, 18H, (CH_3_)_2_N^+^), 3.23–3.29 (m, 8H, NHCH_2_CH_2_CH_2_N^+^), 3.52 (m, 8H, NHCH_2_CH_2_CH_2_N^+^), 3.93 (d, ^3^*J*_HH_ = 5.8 Hz, 8H, OCH_2_CH_3_), 4.07 (m, 8H, N^+^CH_2_CO), 4.13 (m, 8H, NHCH_2_CO), 4.39 (d, ^2^*J*_HH_ = 13.6 Hz, 2H, OCH_2_C(O)), 4.49 (s, 2H, OCH_2_C(O)), 4.51 (s, 2H, OCH_2_C(O)), 4.78 (d, ^2^*J*_HH_ = 13.6 Hz, 2H, OCH_2_C(O)), 7.01 (d, ^4^*J*_HH_ = 2.4 Hz, 2H, ArH), 7.65 (d, ^4^*J*_HH_ = 2.4 Hz, 2H, ArH), 7.68 (s, 2H, ArH), 7.75 (s, 2H, ArH), 8.31 (br.s, 2H, NHCH_2_CH_2_CH_2_N^+^), 8.41 (br.s, 1H, NHCH_2_CH_2_CH_2_N^+^), 8.50 (br.s, 1H, NHCH_2_CH_2_CH_2_N^+^), 9.05. (m, 4H, CONHCH). ^13^C NMR (DMSO-*d_6_*, δ, ppm): 14.1, 22.6, 30.7, 31.0, 33.8, 35.5, 40.8, 51.3, 60.8, 61.8, 62.7, 72.6, 119.5 (^1^*J*_CF_ = 322 Hz), 126.4, 127.1, 127.6, 128.1, 133.7, 134.0, 135.1, 135.4, 145.3, 145.7, 146.5, 157.2, 159.4, 163.7, 166.9, 168.0, 169.1. Elemental analysis. C_100_H_144_F_24_N_16_O_36_S_12_ C, 40.21; H, 4.86; F 15.26, N, 7.50; S, 12.88 found: C, 39.26; H, 4.46; F 15.08, N, 6.95; S, 12.49. HRMS (ESI), *m/z*: calculated for: 466.2370 [M–4 NTf_2_^−^]^4+^, 715.2898 [M–3 NTf_2_^−^]^3+^, 1212.8936 [M–2 NTf_2_^−^]^2+^; found: 466.2357 [M–4 NTf_2_^−^]^4+^, 715.2875 [M–3 NTf_2_^−^]^3+^, 1212.8914 [M–2 NTf_2_^−^]^2+^. FTIR ATR (ν, cm^−1^):1094 (C–O–C), 1674 (C=O), 3207 (N–H).

#### 3.5.3. 5,11,17,23-Tetra-*tert*-butyl-25,26,27,28-tetrakis{*N*-[3′-(dimethyl{[(*S*)-ethoxycarbonylbenzylmethyl]aminocarbonylmethyl}ammonio)propyl]aminocarbonylmethoxy}-2,8,14,20-thiacalix[4]arene tetra[bis(trifluoromethylsulfonyl)imide] in *partial cone* Conformation (**19**)

Yield: 0.125 g (95%). M.p. 55 °C. ^1^H NMR (DMSO-*d_6_*, δ, ppm, *J*/Hz): 1.00 (s, 18H, (CH_3_)_3_C), 1.10 (t, ^3^*J*_HH_ = 7.1 Hz, 12H, CH_3_CH_2_O), 1.27 (s, 9H, (CH_3_)_3_C), 1.28 (s, 9H, (CH_3_)_3_C), 1.86 (m, 8H, NHCH_2_CH_2_CH_2_N^+^), 2.90–2.95 (m, 8H, CH_2_Ph), 3.02 (s, 6H, (CH_3_)_2_N^+^), 3.08 (s, 18H, (CH_3_)_2_N^+^), 3.13–3.24 (m, 8H, NHCH_2_CH_2_CH_2_N^+^), 3.42–3.45 (m, 8H, NHCH_2_CH_2_CH_2_N^+^), 4.04–4.07 (m, 16H, CH_3_CH_2_O, N^+^CH_2_CO), 4.40 (d, ^2^*J*_HH_ = 13.5 Hz, 2H, OCH_2_C(O)), 4.49 (s, 2H, OCH_2_C(O)), 4.52 (s, 2H, OCH_2_C(O)), 4.56–4.61 (m, 4H, NHCHCO), 4.79 (d, ^2^*J*_HH_ = 13.5 Hz, 2H, OCH_2_C(O)), 7.02 (d, ^4^*J*_HH_ = 2.4 Hz, 2H, ArH), 7.21–7.28 (m, 20H, Ph), 7.65 (d, ^4^*J*_HH_ = 2.4 Hz, 2H, ArH), 7.67 (s, 2H, ArH), 7.75 (s, 2H, ArH), 8.28 (br.s, 2H, NHCH_2_CH_2_CH_2_N^+^), 8.40 (br.s, 1H, NHCH_2_CH_2_CH_2_N^+^), 8.48 (br.s, 1H, NHCH_2_CH_2_CH_2_N^+^), 9.09. (m, 4H, CONHCH). ^13^C NMR (DMSO-*d_6_*, δ, ppm): 13.9, 22.6, 30.7, 31.0, 33.8, 35.5, 36.6, 51.3, 53.6, 61.0, 61.7, 62.6, 72.6, 119.5 (^1^*J*_CF_ = 322 Hz), 126.3, 126.8, 127.7, 128.4, 129.2, 133.7, 134.1, 135.2, 135.4, 136.6, 145.4, 145.7, 146.6, 157.3, 163.1, 168.0, 168.8, 170.7. Elemental analysis. C_128_H_168_F_24_N_16_O_36_S_12_ C, 45.93; H, 5.06; F, 13.62; N, 6.69; S, 11.49; found: C, 45.79; H, 5.00; F, 13.35; N, 6.43; S, 11.47. HRMS (ESI), *m/z*: calculated for: 556.5348 [M–4 NTf_2_^−^]^4+^, 835.3524 [M–3 NTf_2_^−^]^3+^; found: 556.5206 [M–4 NTf_2_^−^]^4+^, 835.3283 [M–3 NTf_2_^−^]^3+^. FTIR ATR (ν, cm^−1^): 1096 (COC), 1668 (C=O), 3064 (N–H).

#### 3.5.4. 5,11,17,23-Tetra-*tert*-butyl-25,26,27,28-tetrakis{*N*-[3′-(dimethyl{[(*S*)-ethoxycarbonylbenzylmethyl]aminocarbonylmethyl}ammonio)propyl]aminocarbonylmethoxy}-2,8,14,20-thiacalix [4]arene tetra[bis(trifluoromethylsulfonyl)imide] in *1,3-alternate* Conformation (**21**)

Yield: 0.129 g (98%). M.p. 75 °C. ^1^H NMR (DMSO-*d_6_*, δ, ppm, *J*/Hz): 1.12 (t, ^3^*J*_HH_ = 7.1 Hz, 12H, CH_3_CH_2_O), 1.19 (s, 36H, (CH_3_)_3_C), 1.89 (m, 8H, NHCH_2_CH_2_CH_2_N^+^), 2.91–2.96 (m, 8H, CH_2_Ph), 3.08 (s, 24H, (CH_3_)_2_N^+^), 3.15 (m, 8H, NHCH_2_CH_2_CH_2_N^+^), 3.43 (m, 8H, NHCH_2_CH_2_CH_2_N^+^), 3.99 (s, 8H, OCH_2_CO), 4.03–4.07 (m, 16H, CH_3_CH_2_O, N^+^CH_2_CO), 4.59 (m, 4H, NHCHCO), 7.23–7.28 (m, 20H, Ph), 7.59 (s, 8H, ArH), 8.00 (br.s, 4H, NHCH_2_CH_2_CH_2_N^+^), 9.07 (d, ^3^*J*_HH_ = 7.6 Hz, 4H, CONHCH). ^13^C NMR (DMSO-*d_6_*, δ, ppm): 13.9, 22.6, 30.8, 33.9, 35.8, 36.6, 51.2, 53.6, 61.0, 61.7, 62.5, 71.0, 119.5 (^1^*J*_CF_ = 322 Hz), 126.7, 127.6, 128.4, 129.2, 133.1, 136.6, 146.1, 157.2, 163.1, 167.4, 170.7. Elemental analysis. C_128_H_168_F_24_N_16_O_36_S_12_ C, 45.93; H, 5.06; F, 13.62; N, 6.69; S, 11.49; found C, 45.22; H, 5.05; F, 13.53; N, 6.09; S, 11.09. HRMS (ESI), *m/z*: calculated for: 556.5348 [M–4 NTf_2_^−^]^4+^, 835.3524 [M–3 NTf_2_^−^]^3+^; found: 556.5198 [M–4 NTf_2_^−^]^4+^, 835.3278 [M–3 NTf_2_^−^]^3+^. FTIR ATR (ν, cm^−1^): 1093 (COC), 1669 (C=O), 3064 (N–H).

## 4. Conclusions

Novel macrocyclic quaternary ammonium ILs containing amino acid fragments (glycine and *L*-phenylalanine) based on *p-tert*-butylthiacalix[4]arene in *cone*, *partial cone*, and *1,3-alternate* conformations were synthesized. The melting temperature of the obtained ILs was found in the range of 50–75 °C. Replacement of the bromide anion with bis(trifluoromethylsulfonyl)imide led to a decrease in the melting point by 39–55 °C. The ILs in *partial cone* conformation had the lowest melting points among all stereoisomers. Thermal stability of the obtained macrocyclic ILs was determined via thermogravimetry and differential scanning calorimetry. The onset of decomposition of the synthesized compounds was established at 305–327 °C. The obtained results can be applied to the design of sensor systems capable for target substrate recognition. These compounds can also be used as synthetic biomimetic models of oligo- and polypeptides.

## Data Availability

The data presented in this study are available in Appendix A.

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
