# Peer review of "Macrocyclic Ionic Liquids with Amino Acid Residues: Synthesis and Influence of Thiacalix[4]arene Conformation on Thermal Stability"

_molecules, 2022, doi:10.3390/molecules27228006_

Round 1
Reviewer 1 Report
Reviewer comments (molecules-2037692):
I assessed the manuscript entitled “Macrocyclic ionic liquids with amino acid residues: synthesis and influence of thiacalix[4]arene conformation on thermal stability” submitted by Pavel Padnya , and Ivan Stoikov et al. and found that the mentioned work is interesting based on the influence of thiacalix[4]arene conformation on thermal stability.
Some of the important observations were described here:
--The abstract should be revised and explores the results of the finding of the designed work. There are missing synthesis compounds with the used methodology.
--Scheme 2 should be revised with high resolutions.
--The molecular weight should be checked properly and explain the halogen pattern with compound 6 based on mass spectra.
-- If possible, the application should be discovered in the results and discussion/introduction part.
--Furthermore, I advise authors to find the latest references based on ionic liquids and their applications and the last two years' references and explore it in introduction part.
-- The authors should go for careful proofreading to eliminate (a) grammatical errors; (b) many typos; and also (c) remove unnecessary information and description.
In my opinion, this manuscript can’t be accepted in its current form but it may be accepted after minor revision.
Author Response
Reviewer comments (molecules-2037692):
I assessed the manuscript entitled “Macrocyclic ionic liquids with amino acid residues: synthesis and influence of thiacalix[4]arene conformation on thermal stability” submitted by Pavel Padnya , and Ivan Stoikov et al. and found that the mentioned work is interesting based on the influence of thiacalix[4]arene conformation on thermal stability.
In my opinion, this manuscript can’t be accepted in its current form but it may be accepted after minor revision.
Response:
Dear Reviewer! Thank you very much for carefully reading and reviewing our paper! The manuscript has been revised as suggested. We have tried our best to follow the reviewers’ suggestion.
Some of the important observations were described here:
--The abstract should be revised and explores the results of the finding of the designed work. There are missing synthesis compounds with the used methodology.
Response:
The abstract has been rewritten.
--Scheme 2 should be revised with high resolutions.
Response:
Scheme 2 has been revised in high resolutions.
--The molecular weight should be checked properly and explain the halogen pattern with compound 6 based on mass spectra.
Response:
We apologize for the incorrect calculation of the molecular weight of some compounds. The mass spectra data were checked and corrected.
-- If possible, the application should be discovered in the results and discussion/introduction part.
Response:
Suggested practical applications of the results obtained have been added at the end of Part 2.2.
--Furthermore, I advise authors to find the latest references based on ionic liquids and their applications and the last two years' references and explore it in introduction part.
Response:
Several literature references about ionic liquids and their applications have been added.
-- The authors should go for careful proofreading to eliminate (a) grammatical errors; (b) many typos; and also (c) remove unnecessary information and description.
Response:
The text of the manuscript has been rechecked. Found errors and typos corrected.
Reviewer 2 Report
This manuscript studies a series of novel thiacalix[4]arenes based ammonium ionic liquids (ILs) containing amino acids and their thermal stability. The conformation of the macrocycle has an effect on the melting point of ILs, and the melting point changes between 50‒75 °C with the change of conformation. The stability of macrocyclic ILS was tested in thermogravimetric analysis. Furthermore, the thermally stable of L-phenylalanine residues are 3‒19 °C less than same glycine containing derivatives. This work is interesting. However, the following issues should be addressed before the paper is considered suitable for the publication in molecules.
1. There are some grammatical mistakes in the full text, such as, on page 7, 3.4. General procedure of the compounds 10–15 synthesis, “tetrabromide in cone conformation (11)” should be “tetrabromide in cone conformation (11)”.
2. In introduction, the author says “low biocompatibility and complexation selectivity limit the practical application of these compounds.” Does the compounds in this study have biocompatibility?
3. Table 1. lists the chemical shift values of a series of compounds, but the experimental conditions of the NMR test are not written, please complete this part.
4. This paper mainly studies the thermal stability through thermogravimetric analysis (TG), but the experimental results lack reliability. Please supplement differential scanning calorimeter (DSC) experiments.
5. In introduction, the authors mentioned “there is a direction for obtaining ILs based on supramolecular platforms, e.g. crown ethers, pillararenes, and (thia)calixarenes.” In order to support this statement, the following recently published important related papers should be cited: Chem. Soc. Rev. 2021, 50, 2839; Mater. Today Chem. 2022, 24, 100919.
Author Response
This manuscript studies a series of novel thiacalix[4]arenes based ammonium ionic liquids (ILs) containing amino acids and their thermal stability. The conformation of the macrocycle has an effect on the melting point of ILs, and the melting point changes between 50‒75 °C with the change of conformation. The stability of macrocyclic ILS was tested in thermogravimetric analysis. Furthermore, the thermally stable of L-phenylalanine residues are 3‒19 °C less than same glycine containing derivatives. This work is interesting. However, the following issues should be addressed before the paper is considered suitable for the publication in molecules.
Response:
Dear Reviewer! Thank you very much for carefully reading and reviewing our paper! The manuscript has been revised as suggested. We have tried our best to follow the reviewers’ suggestion.
- There are some grammatical mistakes in the full text, such as, on page 7, 3.4. General procedure of the compounds 10–15 synthesis, “tetrabromide in cone conformation (11)” should be “tetrabromide in cone conformation (11)”.
Response:
The text of the manuscript has been rechecked. Found errors and typos corrected.
- In introduction, the author says “low biocompatibility and complexation selectivity limit the practical application of these compounds.” Does the compounds in this study have biocompatibility?
Response:
It is known that the introduction of biologically active molecules into the (thia)calixarene macrocyclic platform leads to a decrease in the toxicity of target compounds. We hypothesized that combining the thiacalixarene platform, ammonium ionic liquids, and amino acid residues would result in compounds with low toxicity and biocompatibility. In this study, we focused on the synthesis and the study of the physicochemical properties of the macrocyclic ionic liquids with amino acid fragments. The study of toxicity and biocompatibility of the obtained compounds is planned further.
- Table 1. lists the chemical shift values of a series of compounds, but the experimental conditions of the NMR test are not written, please complete this part.
Response:
The experimental conditions of the NMR experiments have been added.
- This paper mainly studies the thermal stability through thermogravimetric analysis (TG), but the experimental results lack reliability. Please supplement differential scanning calorimeter (DSC) experiments.
Response:
The DSC experiments data have been added. Please see Part 2.2 and Figure S34.
- In introduction, the authors mentioned “there is a direction for obtaining ILs based on supramolecular platforms, e.g. crown ethers, pillararenes, and (thia)calixarenes.” In order to support this statement, the following recently published important related papers should be cited: Chem. Soc. Rev. 2021, 50, 2839; Mater. Today Chem. 2022, 24, 100919.
Response:
Recommended literature references have been added.